# Multi-magnification networks for deformable image registration on histopathology images

Oezdemir Cetin[1], Yiran Shu[1], Nadine Flinner[2], Paul Ziegler[2], Peter Wild[2], and Heinz Koeppl[(✉)1]

[1] Department of Electrical Engineering and Information Technology, Technische Universität Darmstadt, Germany
[2] Senckenberg Institute of Pathology, University Hospital Frankfurt, Germany
Dr. Senckenbergisches Institut für Pathologie, Universitätsklinikum Frankfurt, Germany
heinz.koeppl@tu-darmstadt.de

**Abstract.** We present an end-to-end unsupervised deformable registration approach for high-resolution histopathology images with different stains. Our method comprises two sequential registration networks, where the local affine network can handle small deformations, and the non-rigid network is able to align texture details further. Both networks adopt the multi-magnification structure to improve registration accuracy. We train the proposed networks separately and evaluate them on the dataset provided by the University Hospital Frankfurt, which contains 41 multi-stained histopathology whole-slide images. By comparing with methods using the single-magnification structure, we confirm that the proposed multi-view architecture can significantly improve the performance of the local affine registration algorithm. Moreover, the proposed method achieves high registration accuracy of contents at the cell level and is potentially applicable to other medical image alignment tasks.

**Keywords:** Histopathological image · Affine transformation · Non-rigid registration · Unsupervised Learning · Multi-Magnification Network.

## 1 Introduction

Histopathological whole slide images, i.e., digital tissue slides produced by scanning conventional glass slides under high-resolution microscopy, are vital for modern histopathology analysis [15]. Standard whole slide images employ the pyramid structure to support different resolutions, making it easy for pathologists to observe by zooming. Each layer of the pyramid corresponds to a resolution level, with the bottom being the highest resolution information. In general, histopathologists utilise various staining techniques based on chemical features of the tissue, e.g. Hematoxylin-Eosin (H&E), periodic-acid Schiff (PAS) or elastic-van Gieson (EvG). In addition, antibody-mediated visualization of specific proteins, termed immunohistochemistry, is widely used in modern histopathology. As tissue specimens are prepared by approx. 3 µm-thin cuts each specimen represents an unique sample and slides obtained from directly adjacent tissue differ

slightly in their morphology. Even when the same tissue slide is used for multiple staining, e.g. by bleaching and re-staining, shifts and/or deformations inevitably occur. These digital multi-stained histopathology images that are not aligned accurately pose obstacles to the diagnosis or further processing, thus need to be registered first.

Image registration is the process of matching two images geometrically so that corresponding coordinate points in both images correspond to the same physical region of the scene being imaged [21]. Biomedical image registration constitutes one of the key research areas for medical analysis that has been extensively studied. Traditional registration methods search for spatial transformation that brings the defined similarity metric to be optimum by an iterative optimization algorithm [1]. Nevertheless, the superiority of accuracy and robustness of classical approaches come at the cost of time, which becomes the main bottleneck in archiving desirable performance for practical applications. With the revival of deep learning, attempts have been made to develop learning-based approaches to implement faster registration, which can be grouped into three main categories [6,5]: ($i$) deep iterative registration, which follows the framework of traditional methods but instead adopts similarity metrics learned by deep neural networks [16,17], ($ii$) supervised transformation prediction, utilizing the known ground truth transformations to define the cost function [9,18], ($iii$) Unsupervised transformation prediction, where a spatial transformation network is applied to calculate the error of the given metric(s) with an appropriate regularization term [2,19]. The first class of methods inherits the time-consuming drawback of conventional approaches due to the iterative process, whereas the supervised training requires a large amount of data with annotations. In contrast, unsupervised transformation approaches produce the supervisory signals required for training directly by data and can achieve real-time registration during prediction. Therefore, we focused on the unsupervised methods in this work.

An obstacle to applying learning-based methodologies to histopathology images concerns their ultra-high resolution. Some studies have resampled images down to an acceptable memory limit before deformation estimation [15,3]. However, such detailed information as the cell morphological structure is almost impossible to observe on low-resolution images, becoming a key hamper in improving alignment accuracy. An alternative solution is to perform registration on smaller patches [12,8]. The shortcoming of this approach is the irreversible loss of neighboring information when splitting the images, resulting in the narrow field-of-view. In this work, we propose two deep multi-magnification network architectures for patch-based affine and non-rigid registration. The proposed local affine algorithm can effectively deal with imperceptible collective shifts of cell nuclei in the low-resolution pattern, and non-rigid registration is able to align further the cell components that are slightly altered in the morphological structure. We train the presented networks unsupervised and yield higher registration accuracy than the methods using only ordinary single-magnification networks. The result reaches precise alignment at the cellular level under the maximum

resolution of histopathology WSIs, which significantly contributes to the manual/automatic pathological diagnosis on the differently stained tissue sections.

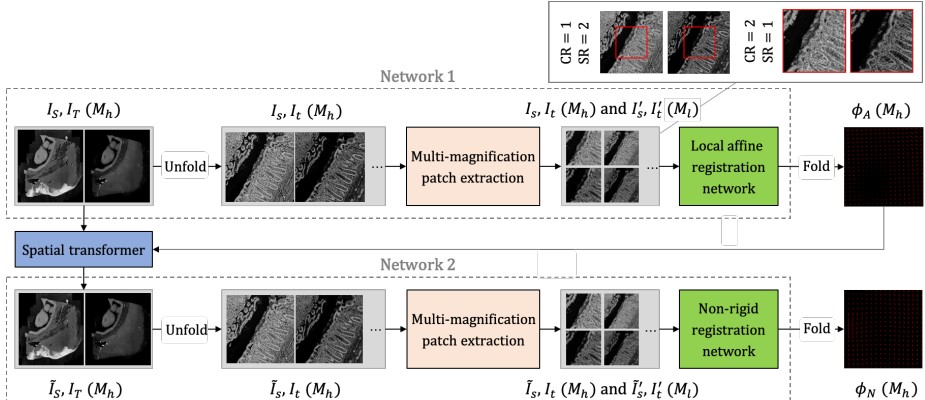

**Fig. 1.** Overview of the proposed algorithm: Both networks take as input concatenated patches $I_s$, $I_t$ for $M_h$ (high) magnification, and concatenated patches $I_s{}'$, $I_t{}'$ for $M_l$ (low) magnification. An example in the upper right corner illustrates the construction process of a patch set, where the cropping rate (CR) and sampling factor (SR) used to build patches for each magnification level are given. The red boxes denote the corresponding regions at different magnifications.

## 2    Methods

Let $I_S$, $I_T$: $\Omega \rightarrow \mathbb{R}$ represent the whole slide source and target images, defined in the spatial domain $\Omega \subset \mathbb{R}^d$, where $d$ denotes ($d = 2$ in this study) spatial dimensionality of the given data. Similarly, $I_s$, $I_t$: $\omega \rightarrow \mathbb{R}$ with $\omega \subset \Omega$ represent the patch-wise source and target images, extracted from $I_S$ and $I_T$. Assuming that the image pairs to be registered are pre-aligned well, we aim to find two deformation fields $\phi_A, \phi_N$: $\Omega \rightarrow \Omega$ to deform the source image such that:

$$I_S(\phi_N \circ \phi_A(x)) \approx I_T, \forall x \in \Omega. \tag{1}$$

Here "$\circ$" represents the composition of deformations and $I(\phi)$ indicates $I$ deformed by $\phi$. The deformations $\phi_A$, $\phi_N$ are defined as a patch-wise affine deformation and a pixel-wise non-rigid deformation, respectively. They are obtained by aggregating the local deformations $\phi_a^p$, $\phi_n^p$ : $\omega \rightarrow \omega$ of image patches $(I_s, I_t)$ extracted from $(I_S, I_T)$, where $p$ indicates the index of the patch on the whole slide image. Two convolutional neural networks $f_a$ and $f_n$ are used to realize the affine registration $\phi_a = f_a(I_s, I_t)$ and non-rigid registration $\phi_n = f_n(I_s(\phi_a), I_t)$, respectively.

An affine registration network is leveraged to learn the affine transformation $\phi_a := Tx$, where $T \in \mathbb{R}^{d \times m}$ with $m = d + 1$. Next, the affinely registered images are fed into the non-rigid registration network to learn the displacement field $u(x)$ with $\phi_n := x + u(x)$, which represents the displacements for $\forall x \in \omega$ in the vertical and horizontal directions.

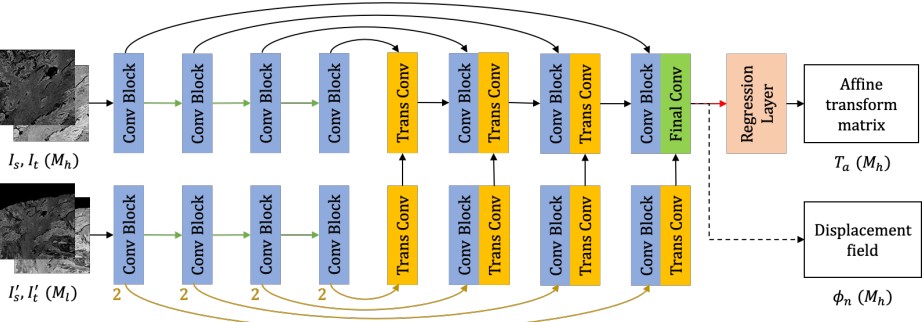

**Fig. 2.** Architecture of the local affine and non-rigid registration networks: $ConvBlock$ includes two sets, each consisting of a $3 \times 3$ convolution layer with group normalization (GN), activated by PReLU. $TransBlock$ comprises a $2 \times 2$ transposed convolution layer with a stride of 2 followed by GN and PReLU activation. The green and red arrows indicate maximum pooling and average pooling, respectively. The center cropping operations are denoted by brown arrows with the cropping rates written in brown. Other blocks are described in the text.

The input of both networks is a set of image patches with different magnifications, providing multiple field-of-views to the networks. Fig.1 offers an overview of the proposed registration algorithm for the case of two magnification levels. The strategy adopted for extracting multi-magnification patches in this work is described as follows: In a multi-magnification set, all other patches are obtained by center-cropping the base image with different cropping rates. Then, the patches are downsampled with the corresponding sampling factors to uniform the patch size. The downsampled base image is the one with the lowest magnification level in the set. Registration networks take the patch set as input and predict the local affine transform matrix/displacement field corresponding to the patch with the highest magnification level, as details described in the next section. According to equation 1, the final deformation for the given images $I_S$ and $I_T$ is obtained by composing the folded $\phi_A$ and $\phi_N$.

### 2.1   Network Architectures

The proposed networks are inspired from [7], which contains multiple magnification layers that obtain more information from different field-of-views. Considering that the architectures of both networks are quite similar, they are shown

in one figure for brevity, as visualized in Fig.2. The concatenation of the high-magnification patches $I_s$ and $I_t$ is fed into the target magnification layer based on the U-Net [13], to extract the higher magnification feature maps. During reconstruction, these feature maps are concatenated with the corresponding lower magnification feature maps extracted from the lower-magnification patches $I_s{}'$ and $I_t{}'$ in another magnification layer. To limit the usage of feature maps from cropped boundary areas in a wide field-of-view, the lower magnification feature maps are center-cropped with a given cropping rate followed by up-sampling utilizing transpose convolution to match the size.

In the local affine network, $Final Block$ has the same structure as $ConvBlock$ but a stride of 2, followed by an adaptive average pooling layer. The reconstructed feature maps are transformed into six numeric parameters through a fully-connected layer and then rearranged into the resulting affine transform matrix $T$ in the regression layer. Whereas, in the non-rigid network, the reconstructed feature maps are compressed utilizing $Final\ Block$, a stack of a $3 \times 3$ and a $1 \times 1$ convolution layer, into two-channel displacement field $u(x)$.

## 2.2 Loss Function

Assume that $\phi : \omega \rightarrow \omega$ is the local deformation field estimated by networks with image patches $I_s$ and $I_t$ as input, the loss function can be described as

$$\mathcal{L}\left(I_s, I_t, \phi\right) = \mathcal{L}_S\left(I_s(\phi), I_t\right) + \lambda\mathcal{L}_R\left(\phi\right), \tag{2}$$

where the first term $\mathcal{L}_S$ measures the similarity between the warped source and the target patches, and $\mathcal{L}_R$ is a regularization term considered only in the non-rigid network. Parameter $\lambda$ controls the trade-off between these two terms as a hyperparameter in the training process.

We choose the normalized cross-correlation (NCC) [10] as the similarity metric $\mathcal{L}_S$. Let $I_1$, $I_2$ be two images then this similarity can be computed as

$$NCC(I_1, I_2) = \frac{1}{N-1} \sum_{x\in\omega} \frac{(I_1(x) - \bar{I}_1)(I_2(x) - \bar{I}_2)}{\sigma_{I_1}\sigma_{I_2}}, \tag{3}$$

where $N$ indicates the number of non-zero pixels, $\bar{I}$ and $\sigma_I$ represent the mean and standard deviation of the intensities in image $I$, respectively. The negative normalized cross-correlation (NCC) is used in training to minimize the loss function, while a higher NCC value corresponds to a higher similarity between images.

Under the intuition that a desirable deformation field should not vary too much between nearby points, the curvature regularization [4] is used to constrain the geometric smoothness of the displacement field $\phi$ predicted by the non-rigid network, i.e.,

$$\mathcal{L}_R(\phi) = \sum_{x\in\omega} \| \nabla\phi(x) \|^2 . \tag{4}$$

## 3    Experiments

The University Hospital Frankfurt (UKF) provided the images used in this study to evaluate the proposed algorithm, with clinical data removed and completely anonymized. The UKF dataset comprises two parts: The first part offers 36 histopathological WSIs, where every two images are from the same tissue section, respectively stained with H&E and IHC-CD8. The second part consists of 5 WSIs obtained from two staining experiments in which multiple staining was performed on the tissue slides from one tissue in different orders. All WSIs are provided as .mrxs files with a unified specification. Each of them contains images at nine resolutions with a downsampling factor of 2, where the full resolution exceeds $180k \times 90k$ pixels in size. We generated 18 and 5 image pairs respectively from two parts of the UKF dataset for training and evaluation. The experiment details are presented next.

### 3.1    Experimental Settings

**Data Preprocessing**  We removed large background areas in the raw data by a boundary detection algorithm and then converted them into single-channel grayscale images. The rigid alignment method derived from [20] was adopted to handle the large misalignment of the image pairs.

**Technical Details**  The proposed algorithm was implemented by modifying and extending the DeepHistReg framework [20]. Unsupervised methods were trained on the resolution-level 4 images whose size varies from $3k$ to $7k$ pixels in one dimension. The images are split into overlapping patches, followed by extracting $224 \times 224$ patches of different magnification levels as the input to the networks. We trained both presented networks with a batch size of 4 using Nvidia Tesla P100 (PCIe). The Adam optimizer with an initial learning rate of $1e-3$ and a decay rate of 0.95 was adopted to update the network parameters. The constraint coefficient $\lambda$ for the non-rigid network training was chosen to be 60.

**Baseline Methods**  We built two single-magnification networks for local affine and non-rigid registration as the baseline models for comparison. The architecture of both networks inherited the target magnification layer of the corresponding multi-amplification network with some adaptations. The training settings were the same as the proposed methods.

### 3.2    Evaluation Metrics

We quantified the registration accuracy by several similarity metrics since no ground truth such as landmarks or segmentation maps are provided for the UKF dataset. Except for the metric NCC used as the objective function during network training, the quality of the deformation fields was also evaluated by the

Mean-Squared-Error (MSE) [11] and the normalized Mutual-Information (NMI) [14], which are respectively defined as

$$MSE(I_1, I_2) = \frac{1}{N} \sum_{x \in \omega} [I_1(x) - I_2(x)]^2 , \tag{5}$$

$$NMI(I_1, I_2) = \frac{2 \cdot H(I_1, I_2)}{H(I_1) + H(I_2)}, \tag{6}$$

where $H$ indicates Shannon's entropy and $H(I_1, I_2)$ represents the dependence of variables (images) $I_1$ and $I_2$.

**Table 1.** Comparison among methods with single/multi-magnification registration networks, containing the average inference time and performance quantified by the similarity metrics NCC, MSE, and NMI (arrows indicate the trend of the increased similarity): The methods are named according to the adopted network architectures, where S/M stands for networks with the single/multi-magnification structure, and A/N denotes the local affine transformation and non-rigid deformation. For example, MASN refers to combining a multi-magnification local affine network and a single-magnification non-rigid network. Besides, an iterative approach is applied based on the presented method, with the number of iterations denoted in parentheses.

| Metric | Initial | SASN | SAMN | MASN | MAMN | MASN(3) |
|---|---|---|---|---|---|---|
| NCC ↑ | 0.6828 | 0.7123 | 0.7060 | **0.7461** | 0.7443 | **0.7728** |
| MSE ↓ | 0.0403 | 0.0376 | 0.0382 | **0.0336** | 0.0338 | **0.0305** |
| NMI ↑ | 0.1670 | 0.1781 | 0.1756 | 0.1952 | **0.1954** | **0.2038** |
| Time (sec) | - | 25.48 | 28.96 | 28.95 | 32.31 | 45.29 |

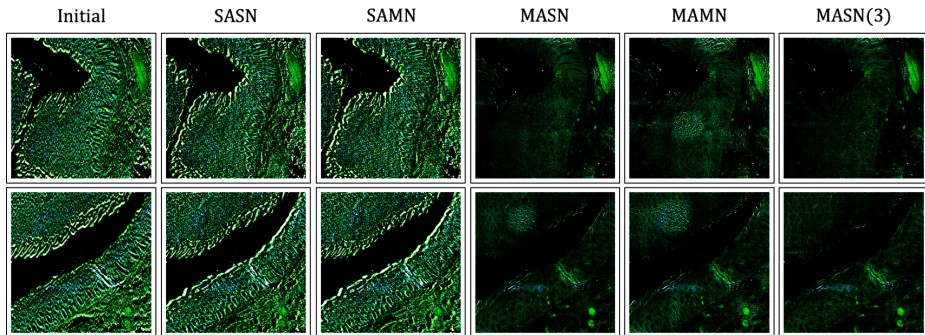

**Fig. 3.** Local subtractions of a high-resolution image pair registered by different methods: The non-overlapping regions appear as fluorescent green due to the nature of stains. For visibility, the contrast/brightness of images has been increased by 50%.

## 4   Results

Table 1 summarizes the overall performance of our proposed algorithm in comparison to approaches containing one or more baseline models. All of them take images pre-aligned by rigid alignment as input.

As shown in Table 1, the proposed multi-magnification structures outperformed the ordinary single-magnification architecture for the local affine algorithm with remarkable benefits, whereas yielding almost no improvement in the performance of the non-rigid network. The increase in runtime due to the multi-magnification structure is not significant compared to the base runtime (SASN). According to the proposed algorithm, the difference in time will decrease exponentially for smaller image pairs. By iterating the prediction on the previous result by the same network, we obtained registration results with significantly higher accuracy.

We upsampled the predicted deformation fields for generating the registered images at a higher resolution. By performing local subtraction between the deformed source and target images, we evaluated the registration performance of different methods at the cellular level, as illustrated in Fig.3. It can be observed that the local affine network improved by the multi-magnification structure is crucial for the enhancement of the overall performance. The cell nuclei can overlap completely in the best cases.

## 5   Discussion and Conclusion

In modern histopathology multiple staining techniques are used to detect specific structures within biological tissues. Each technique highlights different characteristics of the tissue and proper analysis needs to address the spatial distribution of these characteristics. In this context, we developed two novel deep networks with the multi-magnification structure for patch-based image registration, which can learn peripheral information outside the patches as auxiliary information to improve network performance. The presented method is of great importance for biomedical image registration since studies for them can often be performed only on smaller patches due to the large image size. Moreover, the network architectures can be easily expanded with more magnification levels. Nevertheless, this expansion makes little sense since too many field-of-views may instead negatively affect the network performance, especially for cases with no apparent global misalignment.

Our experiments compared the impact of single- and multi-magnification networks on the overall alignment performance by different network combinations. The results revealed that the multi-magnification structure could significantly improve the performance of the patch-based affine registration network. However, it yielded little success on the local non-rigid network. This might mainly attribute to the transformation nature of these two registration methods. The lack of neighboring information can aggravate the estimation error of deformation for the whole image patch region by the local affine approach, while this

error occurs only within the edge region of the image patches in the non-rigid method due to the dense prediction. Therefore, the enhancement of the non-rigid method by the multi-magnification structure was much less evident than that of the local affine approach. Besides, we introduced an iterative approach on the method with the best performance, which further improved the registration accuracy, with an acceptable growth of inference time. The proposed method has the potential to be applicable for other medical image registration tasks.

## Acknowledgement

H.K. acknowledges support from the European Research Council (ERC) with the consolidator grant CONSYN (nr. 773196). O.C. is supported by the Alexander von Humboldt Foundation Philipp Schwartz Initiative.

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
