# OpenReview forum: "Multi-magnification networks for deformable image registration on histopathology images"
_WBIR.info/2022/Workshop/Biomedical_Imaging_Registration — WBIR 2022_

### Official Review · Reviewer_UufW · 2022-02-11

**Rating:** 4
**Confidence:** 4
**Recommendation:** Short Oral

**Deanonymize Review:**

no

**Detailed Comments:**

-

**Paper Type:**

methodological development

**Strengths Weaknesses:**

The paper presents a solution to the bottleneck of whole slide images being too large to register in their entirety for methods based on neural networks. The images are then commonly subdivided into patches, but this results in poor results at patch boundaries. The presented approach tackles this problem using multiple fields of view of patches. It performs local affine and nonrigid registration of slides. The approach is trained and evaluated on clinical datasets of histological slides with multiple stains.

Strengths:
-	An interesting approach that seems to address the problem successfully
-	Potentially useful for other applications
-	Evaluated on clinical data

Weaknesses:
-	Paper is not very clear
-	The evaluation of the method is not correct

My main comment on the paper is that it lacks details and clarity. Even after reading it several times, I failed to fully understand the method. I struggle with the meaning of multi-magnification. In relation to histological images, I take magnification to mean microscopic magnification: viewing/analyzing the image at for instance 10x or 40x. Based on the description in the paper, however, it seems to mean different fields of view at different resolutions. The cropping rate determines the field of view and because all patches are the same size when processed, a larger field of view is downsampled and thus has lower resolution. But the patches all come from the same slide at the same microscopic magnification, if I understand correctly. But then I get confused by the data description which states that the data come at nine resolutions with a downsampling factor of two. How are those resolutions used in combination with the cropping rate and the sampling rates?
If magnification does mean the microscopic magnification of histological slides, I find this an interesting approach because it uses all the information contained in the data at the scales intended. A downside is that it would make it a rather dedicated approach for histology. If the method uses multiple fields of view rather than magnification, it would be an interesting approach because it could apply to several other problems. It is a more general solution
Other information that is lacking is, for instance, what CR and SR are used in the experiments? How many magnification levels are used? How is the nonrigid transformation modelled?
The evaluation of the approach is based on similarity measures other than the one used as a loss function. This is not acceptable. Image registration performance cannot be measured using similarity metrics, because if those metrics could indicate correctly how well the images are aligned, they should have been used as loss function. It is not valid to evaluate with an MSE measure when NCC is required to register the images. The differently stained images are reduced to greyscale and then apparently a linear relationship between the intensities of the images holds, as NCC can be used for registration. But surely there is no identity relationship between the intensities, so MSE is invalid. For proper evaluation is essential that this can be based on an independent measurement, such as landmarks or segmentations.
A downside of the approach is that an interpolation step is required with the use of two networks. In a classic registration approach with multiple transformation types, this is avoided by using the resulting transformations as input for the next registration step.
Ultimately (but not required the workshop paper), the authors should include comparison with classic registration methods, not based on neural networks. The paper does not explain what the advantages are of the deep learning approach over classic ones, because speed is not critical for this particular application. Solving the issues that arise with registering an image in patches is very useful in itself, but for this application the proposed method should achieve better performance than classic methods in other aspects than computing time as well.

---

### Official Review · Reviewer_apMC · 2022-02-12

**Rating:** 4
**Confidence:** 5
**Recommendation:** Short Oral

**Deanonymize Review:**

no

**Detailed Comments:**

Comment:
Quantifying the registration performance with landmarks/segmentation maps would improve the clinical relevance of this work.

Smoothness of the deformation field can be quantified with the std of the Jacobian determinant.

Adding the reference and moving image in figure 3 will help the reader to understand the experimental setting and dataset used in the experiment.

-----------------------------
Overall, the paper is well-written. Although there are a few flaws in the evaluation, I believe the content and scope of this paper are of interest to the WBIR community.


**Paper Type:**

both

**Strengths Weaknesses:**

Summary:
Existing deep learning-based image registration methods for histopathology images are prohibitive by the ultra-high resolution of the images. This paper introduces an unsupervised image registration method for histopathology images. The proposed method consists of a local affine network and a non-rigid network. Both networks share a similar two-stream U-Net like CNN architecture, which takes as input image patches with multiple magnification levels. The method is evaluated on a private dataset with 41 histopathological WSIs and with three similarity measures (NCC, MSE, NMI). Results demonstrate that the multi-magnification architecture is superior to the single-magnification architecture.

------------
Strengths:

Although the multi-magnification architecture is not new in medical image segmentation, combining the architecture with an unsupervised registration framework is a well-grounded effort. It is also good to see the multi-magnification architecture outperforms the common U-Net structure in the context of image registration.

Good reproducibility. The authors provided sufficient details about the models and the source code is made publicly available.

The problem is well-motivated, and this paper proposes a practical solution to address this problem.

The paper is well-written and easy to follow.

------------
Weaknesses:

My major concern is the evaluation metrics used in the experiment. Due to the absence of ground truth deformation/landmarks/segmentation maps of the dataset, the registration performance is quantified with three similarity measures (NCC, MSE, NMI). However, as suggested in [1], evaluating a non-rigid registration method with image similarity is unreliable.

Lack of evaluations on the smoothness of the resulting deformation field. In deformable registration, there is a common trade-off between registration accuracy and smoothness of the deformation field.

---------------------
Reference

[1] Rohlfing, Torsten. "Image similarity and tissue overlaps as surrogates for image registration accuracy: widely used but unreliable." IEEE transactions on medical imaging 31.2 (2011): 153-163.

---

### Official Review · Reviewer_y4P2 · 2022-02-17

**Rating:** 4
**Confidence:** 3
**Recommendation:** Short Oral

**Deanonymize Review:**

no

**Detailed Comments:**

The authors argue well for their aim and the impact of their method, the paper is well-written and nicely presented, the approach seems good though the methodological contribution is arguably relatively minor. Overall, I believe the paper will be of interest to the WBIR audience and I recommend acceptance.

**Paper Type:**

both

**Strengths Weaknesses:**

The paper presents a deformable image registration approach for histopathology images by using a multi-magnification structure to improve accuracy and handle the very high resolution of the histopathology images.

Strengths:
* The paper is well-written and the presentation clear.
* The paper presents a multi-magnifications strategy for registration of histopathology images.
* The experiments and validation show that the method performs well.

Weaknesses:
* As far as I understand, the multi-magnification approach is an adaptation of existing methods ([7] for segmentation). The methodological contribution is therefore not very large.

---

### Decision · Program_Chairs · 2022-02-22

Accept